# Paper-Based Interleukin-6 Test Strip for Early Detection of Wound Infection

**DOI:** 10.3390/biomedicines10071585

**Published:** 2022-07-03

**Authors:** Shin-Chen Pan, Yu-Feng Wu, Yu-Chen Lin, Sheng-Wen Lin, Chao-Min Cheng

**Affiliations:** 1Department of Surgery, Section of Plastic and Reconstructive Surgery, Center of Cell Therapy, National Cheng Kung University Hospital, College of Medicine, National Cheng Kung University, Tainan 704, Taiwan; pansc@mail.ncku.edu.tw; 2International Center for Wound Repair and Regeneration, National Cheng Kung University, Tainan 704, Taiwan; 3Institute of Biomedical Engineering, National Tsing Hua University, Hsinchu 300, Taiwan; fishbee.wu@gmail.com (Y.-F.W.); linseal1009@gmail.com (Y.-C.L.); wenwenlintw@gmail.com (S.-W.L.); 4Division of Plastic Surgery, Department of Surgery, National Taiwan University Hospital, Hsin-Chu Branch, Hsinchu 300, Taiwan

**Keywords:** wound infection, interleukin-6, crp, point-of-care testing, paper-based test strip, spectrum-based optical reader

## Abstract

The early stage of wound infection is always non-specific. Prompt intervention may help to prevent the wound from worsening. We developed a new protocol, based on previous research, that employs a paper-based IL-6 test strip used in combination with a spectrum-based optical reader to detect IL-6 in normal tissue (n = 19), acute wounds (n = 31), and chronic wounds (n = 32). Our data indicated the presence of significantly higher levels of IL-6 in acute wound tissues, but no significant difference in serum CRP. Receiver operating characteristics were used to determine clinical sensitivity and specificity of tissue IL-6 and systemic CRP. The area under the curve values were 0.87 and 0.63, respectively. The cut-off value of 30 pg/mL for IL-6 provided good sensitivity (75.0%) and superior specificity (88.9%). We found a high correlation between the IL-6 test strip and conventional ELISA results (R^2^ = 0.85, *p* < 0.001), and good agreement was also observed according to Bland-Altman analysis. We showed a promising role of tissue IL-6 to help early diagnosis of wound infection when clinical symptoms were non-specific. The advantages of this wound detection protocol included minimal invasiveness, small sample requirements, speed, sample preparation ease, and user-friendliness. This methodology could help care providers quickly clarify wound infection status and implement timely, optimal management.

## 1. Introduction

The early stage of wound infection is always non-specific. A suitable assessment of wound infection status can facilitate optimal therapeutic strategy to promote wound healing. Wound healing can be delayed by multiple factors including tissue necrosis, infection, desiccation, and edema [1]. Infection is one of the most important factors affecting the healing process. Infection can also elicit an inflammatory response to generate biomarkers from local wounds, indicating a potential role of these biomarkers in the early assessment of wound status. A previous study showed that IL-1ß was increased in an inflamed wound with increased bacteria loads [2]. CRP is traditionally used as a biomarker for evaluating infection status. It is a well-known protein expressed in the acute inflammatory phase [3]. Increased CRP levels suggest a developing infection, but the results are not specific. In addition to expression during infection, CRP has reportedly been raised in patients with non-infectious diseases including rheumatoid arthritis [4]. and metastatic melanoma [5], and raised CRP is a risk factor associated with cardiovascular disease [6]. Furthermore, systemic CRP concentrations may not be correlated with local wound status. Wound fluid was reported to contain high levels of pro-inflammatory cytokines including interleukin-1, interleukin-6, and tumor necrosis factor-alpha [7]. Hence, the study of biomarkers such as the interleukins involved in infected wounds might be a more suitable and specific approach for evaluating the infectious state and determining wound prognosis. CRP is activated by interleukin-6 (IL-6) signaling during an inflammatory event [8]. which suggests that IL-6 may be a viable surrogate biomarker for assessing wound infection. IL-6 was reported in the literature as playing an important role in acute and chronic wound inflammation [9,10]. IL-6, as a proinflammatory cytokine, was noted to increase rapidly in the early phase of acute inflammation and steadily decline to a lower level several days later [9]. Although IL-6 was also highly correlated with non-infectious disease [11]. its pro-inflammatory property indicates that it may be useful for predicting the infectious status of wounds.

The measurement of IL-6 via conventional enzyme-linked immunosorbent assay (ELISA) methods is time-consuming and technique dependent. To promote point-of-care wound survey testing for local clinics and achieve the goal of rapid wound assessment, we developed a paper-based test strip to detect biomarkers such as human neutrophil elastase in acute and chronic wounds [12]. Paper-based ELISA technique had the advantages of economy, rapidity, tiny sample demands, and is not inferior to the conventional ELISA method. In this study, we used a new paper-based IL-6 test strip that leveraged via lateral flow immunoassay (LFA) methodology in combination with a powerful spectrometric reader to investigate the predictive value of IL-6 in wound samples for assessing wound infection. We have previously used this test strip to detect IL-6 concentration in patients with influenza [13]. For the sake of comparison, we also investigated systemic CRP levels in this study. Our intent was to demonstrate the suitability of an alternate and convenient approach for assessing wound status in order to provide timely and effective treatment, which could help clinicians with early wound assessment.

## 2. Materials and Methods

### 2.1. Patients and Samples

Samples were harvested from acute wounds, chronic wounds, and normal tissue for analysis. Wound samples were obtained from standard wound debridement surgery. Normal tissue samples were derived from patients undergoing reconstructive surgery. Wound infection was defined as the clinical presentation of wound erythema, local heat, increased exudate, and presence of microorganisms within a wound by laboratory bacteria examination or pathological finding of tissue infection. Serum CRP was examined to survey infection status in most of the studied subjects. To determine IL-6 in sample fluids, tissues were processed as previously described [12]. Briefly, tissue samples were mixed with equivalent volumes of radioimmunoprecipitation assay (RIPA, Millipore, Sigma, St. Louis, MI, USA) buffer to extract protein. Samples were centrifuged at 4000 rpm for 30 min at 4 °C, and then conventional ELISA or IL-6 test strips with a spectrum-based optical reader were used to detect IL-6. A full explanation was made to all and they signed the informed consent form. The study protocol, including participants, sample gathering, and analysis, was approved by the Institutional Review Board of National Cheng Kung University Hospital (No. B-ER-109-238).

### 2.2. CRP

A CRP blood test was used in the clinical laboratory of our hospital using a Beckman Coulter image analyzer. Venous blood samples for CRP assay were collected on the same day as the wound or normal tissues were harvested using the standard procedure in the user guide. Quality control testing was performed according to regular requirements and our standard procedure. CRP was expressed in mg/L. Values were expressed as mean ± standard deviation (SD).

### 2.3. Enzyme-Linked Immunosorbent Assay (ELISA) for IL-6 Measurement in Wound Samples

Commercial ELISA (D6050, R&D systems, USA) kits were used to measure IL-6 values in sample fluids. Each sample was applied in triplicate, and the average data from plates were taken as final values. IL-6 was expressed in pg/mL. Values were expressed as mean ± standard deviation (SD).

### 2.4. Lateral Flow Immunoassay-Based IL-6 Test Strip with Reflectance Spectral Analysis

The IL-6 test strip was developed to employ the LFA methodology. This technique has been described previously [13,14]. The test strip was manufactured by Hygeia Touch Inc. (Taiwan) and used colloidal gold-conjugated anti-IL-6 antibodies (ARG21446, Arigo Biolaboratories Corp., Hsinchu, Taiwan) to identify IL-6 antigens in tissue samples. Sample fluid was applied onto the strip to form an IL-6 and colloidal gold-labeled anti-IL-6-antibody complex. The complex was captured and presented as a colored band on the test line (T-line) of the IL-6 test strip, which could then be evaluated for IL-6 concentration using a spectrum-based optical reader.

The spectrum analyzer (in collaboration with SpectroChip Inc., Taiwan; Taiwan FDA: MD (I)-008090 and U.S. FDA: 3017810861) was developed to detect IL-6 concentrations on the IL-6 test strip. This device provided a continuous spectrum with high resolution (3–5 nm) and captured the IL-6 signal on the aforementioned T line of the strip. The primary reflectance wavelengths of the IL-6 complex were detected between 430 nm and 600 nm and the main reference wavelength was 650 nm. The equation for α value was:α = Reflectance (at 650 nm)/Reflectance (at minimum 430 nm to 600 nm)

The α value was used to indicate IL-6 concentration on the screening test strip. Higher α values indicated higher IL-6 levels on the test strip.

### 2.5. Performance of IL-6 Test Strip

A preliminary test of the IL-6 test strip was undertaken to establish a standard curve using different concentrations of IL-6 recombinant protein with RIPA solution (1 ng/mL, 500 pg/mL, 250 pg/mL, 100 pg/mL, 50 pg/mL, and 0 pg/mL). The 150 μL solution was applied to the test strip and allowed to rest for 10 min. The T-line result was quantified using the spectrum-based optical reader. The reflectance spectra of the IL-6 complex were determined at 540 nm, which was well separated at each concentration [13]. The α value was acquired from the reflectance spectra to construct the IL-6 standard curve of reflectance. After adjusting the α value with each IL-6 concentration, the linear regression had an R^2^ value of 0.907. In this model, the limit of detection (LOD) and limit of quantification (LOQ) were determined as 76.85 pg/mL and 402.71 pg/mL, respectively, thus providing excellent sensitivity for detecting tiny IL-6 concentration in the wound fluids.

### 2.6. Statistical Analysis

The Kruskal-Wallis test was used to analyze the differences in values between normal tissues, acute wounds, and chronic wounds. Values of less than 0.05 were considered statistically significant. The diagnostic abilities of IL-6 and CRP were evaluated using the area under the receiver operating characteristic (ROC) curve (AUC). The correlation and agreement of two different methods such as our paper-based test strip and conventional ELISA were evaluated using the Spearman rank correlation coefficient and a Bland–Altman plot analysis.

## 3. Results

### 3.1. Characteristics of the Study

A total of 82 wound and normal tissue samples (normal tissue: 19, acute wounds: 31, chronic wounds: 32) from 53 patients were harvested for analysis of IL-6 in tissue fluids. Detailed patients’ information is shown in Appendix A. Fifteen patients had serial wound samples collected due to persistent infection. Systemic CRP was investigated in 71 episodes from 42 patients. The summary of patients’ information including infected and non-infected case number in each group, IL-6, and systemic CRP data are shown in Table 1. Infection was defined as the presence of wound erythema, local heat, increased exudate, and presence of microorganisms within a wound by laboratory bacteria examination or pathological finding of tissue infection. Chronic wounds were wounds that did not improve within 6 weeks.

### 3.2. Tissue IL-6 Is a Superior Prognostic Marker to Predict Wound Infection Status

Although CRP is a sensitive infection marker, it seems somewhat non-specific for determining local wound infection status. As with previous research, our data did not support the role of systemic CRP in determining wound infection status. There was no significant difference in CRP levels between acute, chronic, and normal tissue samples (Figure 1A). IL-6, on the other hand, is known as an acute-phase inflammatory marker that releases inflammatory cytokines from activated inflammatory cells and triggers proinflammatory signaling to prevent infection. [15] To further confirm the role of IL-6 in the assessment of local wound status, we evaluated IL-6 values in three different samples using conventional ELISA. The data indicate that IL-6 was highly expressed in acute wound tissues compared to chronic wound and normal tissue samples (Figure 1B), suggesting a promising role for IL-6 as a biomarker for evaluating wound infection.

### 3.3. Predictive Role of IL-6 in Local Wound Status

To establish the diagnostic ability of IL-6 for discriminating wound infection, we examined the receiver operating characteristic (ROC) curve of IL-6 and CRP to determine clinical sensitivity and specificity and examined every possible cut-off value. As shown in Figure 2, the area under the curve (AUC, 95%CI) value for IL-6 was 0.87 (0.79–0.95), compared to 0.63 (0.49–0.76) for systemic CRP (Figure 2A). We calculated several cutoff values, such as 20, 25, and 30 pg/mL of IL-6 level, and found that the sensitivity and specificity were 80.6%, 64.4%, 80.6%, 77.8%, 75.0% and 88.9%, respectively (Figure 2B). According to the measured IL-6 levels, the cut-off value of 30 pg/mL provided an optimal cut-off point to help early diagnosis of wound infection with non-specific symptoms. In addition, ROC and AUC of systemic CRP were also evaluated. The AUC of CRP was 0.63 (0.49–0.76), which did not show the expected results as IL-6 (Figure 2C,D). The CRP data did not suggest a predictive role for wound infection.

### 3.4. IL-6 Test Strip Provides a Reliable and Convenient Tool for Identification of IL-6

We confirmed the role of tissue IL-6, as opposed to systemic CRP, in predicting wound infection via conventional ELISA. However, traditional ELISA procedures are time-consuming and expensive. To develop a novel point-of-care device for clinical wound assessment, we investigated the suitability of an IL-6 test strip used in combination with a spectrum-based optical reader to detect IL-6 concentration in tissue fluids. Sample fluids from normal tissues, acute wounds, and chronic wounds were loaded onto the sample zone of our IL-6 test strip. Visible test results took the form of a colored band on the T-line of the test strip. We observed a significant band shadow on the test strip for acute wound samples compared to the results from normal tissue and chronic wound samples (Figure 3A). To verify the consistency of our IL-6 test strip results with those of conventional ELISA, we selected 15 samples to compare and correlate data. Comparative data indicate a significant correlation between IL-6 test strip results and conventional ELISA method results (R^2^ = 0.85, *p* < 0.001, Figure 3B). A Bland–Altman analysis was applied to validate the agreement between the two measurement types. A 95% limit of agreement (±1.96 standard deviation) was calculated and is displayed here in a scatter plot (Figure 3C). These results suggest high reliability for our IL-6 test strip when used to assess wound status via examination of IL-6 in wound fluids.

## 4. Discussion

Evaluation of the predictive value of specific biomarkers in wound tissue to establish wound prognosis was our goal. Our data suggest a promising role for the evaluation of tissue IL-6, rather than systemic CRP, to assess wound infection status. Our IL-6 test strip used in combination with a spectrum-based optical reader is an ideal tool for the detection of tissue IL-6 as a means of providing timely and rapid analysis of wound infection status.

The systemic CRP data was not specific in our study. CRP has been well studied. It has been identified as an acute-phase protein that is highly expressed in infectious and non-infectious diseases [16]. Its diagnostic potential for wound infection might be restricted due to the non-specific feature. CRP was found not to elevate in venous ulcer wounds [17]. The inflammatory biomarkers in wound fluid can be generated by the systemic response and locally activated wound cells [12]. For this reason, systemic CRP is not strongly associated with local wound infection status. Consistent with this observation, there were two normal tissue samples presented with high CRP levels (> 100 100 pg/mL), further supporting the non-specific role of systemic CRP in wound prognosis. In addition to synthesis by hepatocytes, CRP can be synthesized by different kinds of cells such as smooth muscle cells, macrophages, endothelial cells, lymphocytes, and adipocytes [3]. The multi-source nature of CRP is another reason for its non-specificity. CRP has two isoforms, native CRP and monomeric CRP, which operate in opposing directions [3]. The fact that it is difficult to identify particular CRP isoforms in clinical laboratories may contribute to its loss of diagnostic utility for assessing wound infection status. For these reasons, it remains controversial whether or not systemic CRP is a suitable independent biomarker for wound monitoring without systemic infection. Our finding of irrelevant systemic CRP levels in three different types of wound samples further supports this opinion. Future studies of tissue CRP may provide additional or improved analysis of diagnostic suitability for evaluating wound infection status.

The composition of wound fluid was known to differ from the serum and was recognized as a predictor of wound status [18]. We harvested and centrifuged wound tissue with standard extraction protocol to obtain the tissue fluid. Despite processing in different protocols, a mixture of bacterial products and biomarkers in our wound fluids was consistent with the previous report to have a potential for clinical diagnosis [2]. We detected significantly higher IL-6 levels in acute wounds compared to chronic wounds and normal tissue samples. In response to acute inflammation, IL-6 stimulated many cells such as macrophages, keratinocytes, endothelial cells, and stromal cells to release proinflammatory cytokines and attract leukocytes entering the wound [15]. In non-healing chronic wounds, a low level of IL-6 was detected [19]. Previous reports demonstrated that IL-6 was a good predictor of disease progression in patients with severe acute respiratory syndrome coronavirus 2 [20]. Circulating IL-6 correlated with burn severity [21]. Consistent with this notion, the differential levels of IL-6 between acute and chronic wounds further support the role of IL-6 to give a relevant reflection of the wound status. Circulating IL-6 rather than procalcitonin, WBC, and body temperature had the clinical utility to predict the early stage of sepsis in burned patients [22]. We are committed to finding sensitive quantitative capacities for our IL-6 test strip. The IL-6 cut-off value of 30 pg/mL demonstrated good sensitivity (75.0%) and superior specificity (88.9%), suggesting a potentially ideal approach for wound assessment when clinical wound infection symptoms were unclear. From our observations, 14 of 18 chronic wound samples exhibited higher IL-6 levels (>30 pg/mL), indicating possible wound deterioration.

Procalcitonin, a peptide precursor of calcitonin, has been recognized as a reliable marker in distinguishing bacterial infections when clinical symptoms were not specific [23]. Procalcitonin remained at a low level in non-infected patients but increased rapidly in a variety of infections [24]. A superior diagnostic accuracy of procalcitonin was shown when compared with WBC, ESR, CRP, IL-6, and IL-8 [25]. Although a specific indicator of procalcitonin for determination of infection status was reported, a different study has shown a complex role of increased procalcitonin in non-infectious conditions such as severe trauma, burn, and thyroid cancer and low level in infectious cases [26]. Future studies for procalcitonin test strips may be required to determine if this is feasible for the diagnosis of wound infection.

Early recognition of wound status and support with appropriate management is a basic requirement for patients. Serum infection markers such as ESR, CRP, and blood cell count are not equivalent to wound status. We developed a paper-based diagnostic device that, coupled with a smartphone camera, can help in monitoring the infectious status of wounds, and early initiating intervention of infection control. To the best of our knowledge, there is no commercial detection device available for examining biomarkers in wound tissue. Our paper-based IL-6 test strip detection system to investigate IL-6 in wound fluid may be a promising tool to assist clinical wound evaluation. The benefits of our device include minimal invasiveness, small sample requirements, speed, sample preparation ease, and user-friendliness. Compared to a greater amount of reagent (about 550 μL) and longer processing time required (7–8 h) for conventional ELISA, our technique needed only 15 μL of reagent and an hour of processing. The sensitivity of our IL-6 test strip is comparable to conventional ELISA. This methodology could help care providers quickly clarify wound infection status and implement timely, optimal management, including aggressive debridement and systemic anti-microbial agents. To refine our device for clinical use, several drawbacks of the current study must still be addressed. First, the limitation of our study was a small sample size. Despite a good correlation between IL-6 values and conventional ELISA, only 15 cases are analyzed due to limited sample fluids. More cases are needed to verify our hypothesis. Furthermore, the complex wound environment containing bloody substances may influence our colorimetric results. The substrates with higher specificity to IL-6 should be explored to provide different color changes that can be more easily distinguished from bloody fluids. In conclusion, we assessed the potential of wound-contained IL-6 to predict wound infection. IL-6 concentration was found to be significantly higher in acute wounds, while systemic CRP was not significantly different among test groups. Our study also demonstrates that, in association with clinical assessment, a paper-based IL-6 test strip detection system could provide an easy and convenient tool for early recognition of wound infection.

## Figures and Tables

**Figure 1 biomedicines-10-01585-f001:**
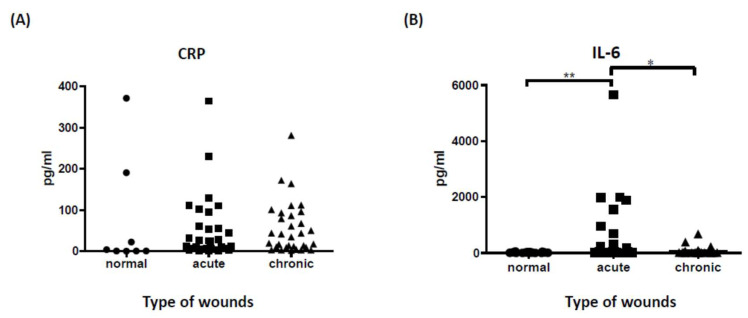
Serum CRP and IL-6 levels detected by conventional ELISA. (**A**) Serum CRP was investigated in 71 episodes of 42 patients between normal tissues, acute, and chronic wounds. There was no statistical difference between different groups (*p* = 0.19, mean± SD). (**B**) IL-6 concentrations in 3 different groups: normal tissue (n = 19), acute wound (n = 31), chronic wound (n = 32). There were statistically significant differences in IL-6 expression levels between acute wound and normal tissue samples (*p* < 0.01) and between acute and chronic wounds (*p* < 0.05). ** *p* < 0.01, * *p* < 0.05, mean ± SD.

**Figure 2 biomedicines-10-01585-f002:**
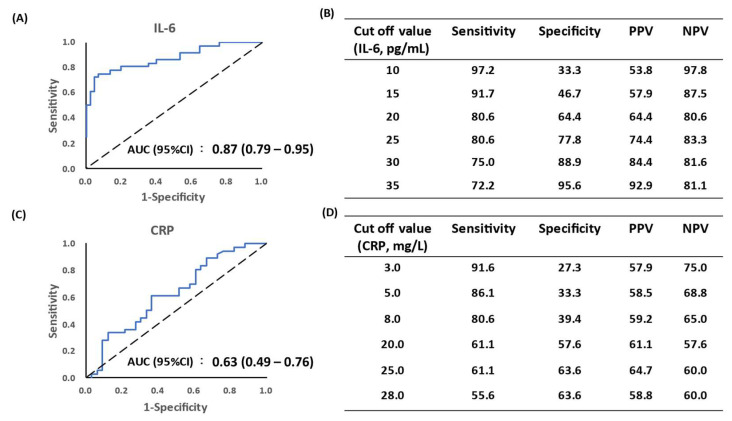
Diagnostic sensitivity and specificity of tissue IL-6 and systemic CRP for wound infection. ROC and AUC were used to evaluate the predictive value of IL-6 and CRP. (**A**) AUC of IL-6 was 0.87. (**B**,**D**) Comparison of sensitivity and specificity of IL-6 and CRP levels based on different cut-off values. (**C**) AUC of CRP was 0.63. ROC, receiver operating characteristic. AUC, area under the curve. PPV, positive predictive value. NPV, negative predictive value.

**Figure 3 biomedicines-10-01585-f003:**
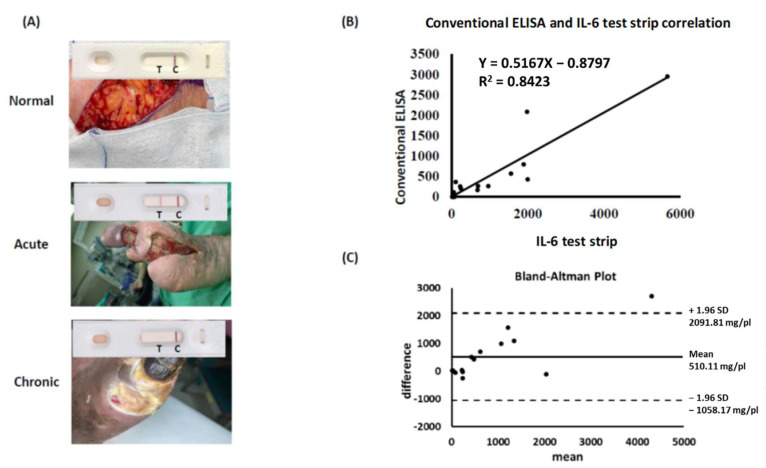
Validation of IL-6 test strip for clinical application. (**A**) Application of IL-6 test strip on clinical samples (normal tissue, acute and chronic wounds). (**B**) Study of the correlation between conventional ELISA and IL-6 test strip results for tissue IL-6 detection (n = 15, *p* < 0.001). (**C**) Bland–Altman analysis of two methods. The differences between conventional ELISA and IL-6 test strips in relation to the mean of the two measurements, n = 15. Dashed lines indicate the limit of agreement (±1.96 SD).

**Table 1 biomedicines-10-01585-t001:** Patients’ information in the acute and chronic wounds and normal tissue.

	Acute	Chronic	Normal
**Case No.**	31	32	19
**Infection (No.)**	22	15	0
**Non-infection (No.)**	9	17	19
**Age (y/o, mean ±** **SD)**	62.8 ± 9.5	64.3 ± 8.9	50.3 ± 11.8
**CRP** **(mg/L, mean ±** **SD)**	51.28 ± 77.79 (IQR: 6.9–61.1)	59.14 ± 63.44 (IQR: 11.7–93)	74 ± 136.87 (IQR: 0.8–106.65)
**IL-6** **(pg/mL, mean ±** **SD)**	522.33 ± 1142.77 (IQR: 19.1–308.88)	51.95 ± 123.47 (IQR: 12.5–36.44)	18.46 ± 14.76 (IQR: 3.49–25.5)

IQR: interquartile range.

## Data Availability

The datasets generated during and/or analyzed during the current study are available from the corresponding author on reasonable request.

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
