# Peer review of "Paper-Based Interleukin-6 Test Strip for Early Detection of Wound Infection"

_biomedicines, 2022, doi:10.3390/biomedicines10071585_

Round 1
Reviewer 1 Report
- The technique of sample preparation should be described in more detail and not referred to in the manuscript by literature.
- Furthermore, what tissue samples are used? Is it just the debris of the debridement? How do you standardize this? What is going on with cross contamination? How do you exclude non inflamed wounds by their bacterial contamination and potential wrong diagnosis?
- For the IL-6-paper test you use wound fluids? How do you get them? What if the wound is dry?
- Table 1: What does the last column mean? Have you double checked this by microbiology swabs? I would further recommend to sort the table by normal, acute and chronic wounds to see potential differences. Otherwise it is a bit confusing in my eyes. Why isn’t there a value for the paper test for all samples?
- Can you comment on the difference of IL-6 positivity as a marker for acute wound infection in contrast to microbial results? The reviewer is not quite clear on the consequence of a positive IL-6 results on the further therapy.
- Line 226ff: again here the question, what is the consequence of therapy, in order to know a high IL-6 level in a wound? What will be done if IL-6 levels even increase, i.e. infection progression?
- Line 234ff: all the advantages have not been properly described in the M&M section. This must be improved significantly. And still, besides the user would know that there is IL-6 in the wound, what will he then do? The reviewer is not clear, if this tool really is cost effective in the sense of the first sentences of the instruction section as well.
Author Response
Point 1: The technique of sample preparation should be described in more detail and not referred to in the manuscript by literature.
Response 1: thanks for the comment, we have modified the sample preparation in material and method section.
Point 2: Furthermore, what tissue samples are used? Is it just the debris of the debridement? How do you standardize this? What is going on with cross contamination? How do you exclude non inflamed wounds by their bacterial contamination and potential wrong diagnosis?
Response 2:
-We obtained the wound tissue by curettage or sharp shaving with knife or scissors, the sample were grossly infected tissue or granulation tissue in acute and chronic wounds and subcutaneous fat tissue in normal tissue group. Debrided wound tissues might contain connective tissue, inflammatory cells or bacteria. Although non-viable tissue was possibly found in our specimen, we have observed the presence of IL-6 in the tissue samples.
-Yes, we have standarized the IL-6 values in our samples by quantification of tissue total protein.
- All harvested sample tissues were encased into sterile tube to prevent cross contamination. Microorganism culture was obtained under standard process, the wound and peri-wound region would be swiped and sterilized with standard disinfection procedures before obtaining wound culturing in order to minimized risk of contamination or mis-diagnosis.
Point 3: For the IL-6-paper test you use wound fluids? How do you get them? What if the wound is dry?
Response 3: It is hard to get proper exudate from wound surface for study. We therefore harvested and centrifuged wound tissue with standard extraction protocol (mentioned in material and method section) to obtain the tissue fluid, rather than directly collected the fluid from the wound surface. Extracted tissue fluid was dripped on test zone of IL-6 strip for analysis.
Point 4: Table 1: What does the last column mean? Have you double checked this by microbiology swabs? I would further recommend to sort the table by normal, acute and chronic wounds to see potential differences. Otherwise it is a bit confusing in my eyes. Why isn’t there a value for the paper test for all samples?
Response 4:
-Thanks for your suggestion. I have reviewed our data and redefined wound infection as “ presence of wound erythema, local heat, increased exudate, and presence of microor-ganisms within a wound by laboratory bacteria examination or pathological finding of tissue infection. After evaluation, 37 cases were enrolled in the infection group cases (normal: 0, acute: 22, chronic: 15), compared to previous 63 infection cases (normal: 0, acute: 31, chronic: 32).
- I have made the new Table 1 to cross classification table and adjusted the order of normal tissue, acute and chronic wounds and added additional patient’s information, microbiology data or pathological findings in supplementary Table 1 to prevent confusion.
- We have tested 15 cases using paper-based IL-6 test strip in wound assessment. Due to limited samples, we can not analyze all samples with both techniques. However, a good correlation of test strip and conventional ELISA was observed in our study. We will collect more cases to verify this hypothesis.
Point 5: Can you comment on the difference of IL-6 positivity as a marker for acute wound infection in contrast to microbial results? The reviewer is not quite clear on the consequence of a positive IL-6 results on the further therapy.
Response 5: The potential role of IL-6 is early detection of wound infection when the clinical symptoms are unclear. Both local wound IL-6 level and microorganism culture results could be a marker of infected wound, but it takes at least 3-5 days for wound culture report and there are some microorganisms known as viable but non-culturable. In contrast, wound IL-6 level could be obtained in short time with our protocol and the prototype of our IL-6 paper stripe could be refined and modifed to coapt with wound evalutation if needed. Therefore, if IL-6 level of the wound is high, which indicates the posibility of acute wound infection, further timely and early wound management including broad-spectrum anti-microorganism agents, surgical debridements, or dressing contained anti-microorganism agents should be applied, rather than just await for the culture report. Like examination of CRP level, assessment of IL-6 levels rather than microbial results may quantiate the infection severity of wounds according to the cutoff values.
Point 6: Line 226ff: again here the question, what is the consequence of therapy, in order to know a high IL-6 level in a wound? What will be done if IL-6 levels even increase, i.e. infection progression?
Response 6: IL-6 is known as a pro-inflammatory cytokine, and in this study, we found that significantly higher level of IL-6 in acute infected wound than normal and non-infected wound. If wound IL-6 is high, which means possibility of severe infection status of the wound, aggressive debridement and local/systemic anti-microbial agents should be used as soon as possible. Fifteen patients have serial wound samples collected in our study. Like use of WBC or systemic CRP to evaluate systemic condition, we can follow the variation of tissue IL-6 levels to help assessment of wound infection status.
Point 7: Line 234ff: all the advantages have not been properly described in the M&M section. This must be improved significantly. And still, besides the user would know that there is IL-6 in the wound, what will he then do? The reviewer is not clear, if this tool really is cost effective in the sense of the first sentences of the instruction section as well.
Response 7:
-Thanks for comments, we have addressed the advantages of our method in the introduction part.
-This tool hopes to early derection the infected wound that should be carefully and promptly treated by the speciallist. Therefore, if the wound IL-6 is high, early consultation or patient referral to the wound specialist should be considered. If the user is the wound specialist, timely wound infeciton control and evaluation should be done as mentioned above, to prevent from wound deterioation or even systemic respose. Furthermore, The signal of IL-6 on the test strip was suggested to be activated by a mobile phone APP for prompt wound determination rather than waiting for established wound infection. It is important for clinicians for early wound assessment.
- We apologize to make confision of the first sentence in the introduction part. We have revised and marked this part.

Reviewer 2 Report
Wound inflammation and infection is one of the healing issues that need to be tackle. This research fits perfectly with the actual needs. There are minor comments though before the manuscript gets Accepted.
- Introduction would benefit from more references to support your statements.
- L 53: Can you please mention which pro-inflammatory cytokine you wanted to highlight?
- L. 84: Do you have any Ethical Approval and a Donor consent on your data? Can you clarify this aspect please? Maybe, you mention “study protocol” you should detail which study, with regards to what?
- l. 105 – 106: Can you write it as an equation instead of a text please?
- Table 1: the goal of your work is to prove that the strip test can be used …. Why most of the results are done with ELISA? What happened to the column of IL-6 results? You did not comment on this aspect.
- What happened tot Sample 82? It’s an acute wound and it has PRC and almost no IL-6?
- l. 188: How did you select the data: blind or double-blind method?
Author Response
Response to Reviewer 2 Comments
Wound inflammation and infection is one of the healing issues that need to be tackle. This research fits perfectly with the actual needs. There are minor comments though before the manuscript gets Accepted.
General comments:
Point 1: Introduction would benefit from more references to support your statements.
Response 1: Thanks for comment. Infection refers to the invasion and multiplication of bacteria or viruses within the body, while inflammation is the body's protective response against infection. We have revised our introduction and also modify the title to clearly define inflammation and infection.
Point 2: L 53: Can you please mention which pro-inflammatory cytokine you wanted to highlight?
Response 2: Thanks for comment, we will study the role of IL-6 in wounds and also have revised and marked the changes in the introduction part.
Point 3: L. 84: Do you have any Ethical Approval and a Donor consent on your data? Can you clarify this aspect please? Maybe, you mention “study protocol” you should detail which study, with regards to what?
Response 3: Thanks for comment, we have Ethical Approval and participants’ informed consent approved by the Institiude Review Borard. More detailed description has been added and marked in the manuscript.
Point 4: T l. 105 – 106: Can you write it as an equation instead of a text please?
Response 4: Thanks for comment, we have revised into an equation.
Point 5: Table 1: the goal of your work is to prove that the strip test can be used …. Why most of the results are done with ELISA? What happened to the column of IL-6 results? You did not comment on this aspect.
Response 5: We have tested 15 cases using paper-based IL-6 test strip in wound assessment. Due to limited samples, we can not analyze all samples with both techniques. However, a good correlation of IL-6 test strip and conventional ELISA was observed in our study. We will collect more cases to verify this hypothesis.
Point 6: What happened to Sample 82? It’s an acute wound and it has CRP and almost no IL-6?
Response 6: Sample 82 is a diabetic foot patient with acute infected wound. We also allert the high CRP and low IL-6 in this patient. Low IL-6 may indicate the remission of infection stauts. By constrast, high CRP may be influenced by unknown medical disease of the patient. The data further support our point that systemic CRP may not a good indicator for wound prediction.
Point 7: l. 188: How did you select the data: blind or double-blind method?
Response 7: We enrolled all participants met the inclusion criteria that agreed to join this study during the study period. The clinical surgeon, that gathered the speciemn, judged the wound was acute or chronic and presense of infection related local findings, was unkown of CRP and IL-6 data. The researcher that managed the specimen and analysis of IL-6, CRP data was also unkown about the clincial data of the patients.

Reviewer 3 Report
In this study the use of IL6 strips was compared to IL6 ELISA as an alternative method for assessing wound status. A high correlation was found between IL6 strips and ELISA. Indeed, “the paper-based IL-6 test strip detection system could provide an easy and convenient tool”. However, conclusions regarding wound status cannot be drawn based on the presented evidence. All acute and chronic wounds were marked as infected. Moreover, IL6 levels were significantly lower in chronic wounds. For proper comparison, non-infected acute wounds should have been included. Non-infected chronic wounds would be a good addition but are more difficult to find.
There are already many studies dedicated to finding biomarkers, and many have studied the use of CRP and IL-6. The present paper only adds another method to detect IL6, it is not new as the authors published this in references 12 &13.
Details regarding patient and wound characteristics are missing: age, comorbidities, etc, and wound erythema, local heat, increased exudate or presence of microorganisms are mentioned but not reported. Other possible biomarkers for infection (such as procalcitonin) were not studied.
Line 38 “other causative factors involved in wound inflammation”: what is meant by causative factors?
Line 57: IL-6 will not be useful for all types of wounds. For example, in burn wounds it will be highly increased in absence of infection. Also, IL6 can be expressed by a variety of cells after a multitude of stimuli. For example skin incision can increase IL6 levels for several hours.
CRP measurement is not included in M&M.
Results
Why include table 1 with information per sample? Why are some CRP results and many IL6 strip results missing? Important information is missing: from which patients were multiple samples taken? 15 patients had serial samples for persistent infection, but which patients and was this repeated measurement taken into account?
There is a significant difference in IL6 between acute and chronic wounds, but all wounds were marked as infected. This indicates that IL6 cannot be used as a marker for infection. Why were CRP results of 2 normal samples >100 pg/ml?
3.2 Why is CRP “unreliable for determining local wound infection status” even when it is sensitive? There is more literature on the limited use of CRP for determining infection. Why not test other biomarkers or CRP/IL6 in plasma?
Line 150-1 “suggesting a promising role for IL-6 as a biomarker for evaluating wound infection”: this is not based on presented results. All acute and chronic wounds were infected, all normal wounds were not. IL6 is not useful for infection in chronic wounds as those values were significantly lower than in acute samples. Non-infected acute wounds were not included, so it is not possible to draw a conclusion on infection status without additional info or samples.
3.3. I assume this paragraph relates to the IL6 ELISA results?
What are the benefits of using strips over ELISA? For the strips a spectrum-based optical reader is needed.
Lin 227-8: “our data also support the hypothesis that differential IL-6 levels could discriminate disease severity.” This is not shown here.
The discussion should be improved and expanded.
Author Response
Response to Reviewer 3 Comments
General comments:
Point 1: -In this study the use of IL6 strips was compared to IL6 ELISA as an alternative method for assessing wound status. A high correlation was found between IL6 strips and ELISA. Indeed, “the paper-based IL-6 test strip detection system could provide an easy and convenient tool”. However, conclusions regarding wound status cannot be drawn based on the presented evidence. All acute and chronic wounds were marked as infected. Moreover, IL6 levels were significantly lower in chronic wounds. For proper comparison, non-infected acute wounds should have been included. Non-infected chronic wounds would be a good addition but are more difficult to find.
Response 1: Thanks for your suggestions. Your point is right. I have reviewed our data and redefined wound infection as “ presence of wound erythema, local heat, increased exudate, and presence of microorganisms within a wound by laboratory bacteria examination or pathological finding of tissue infection. After evaluation, 37 cases were enrolled in the infection group cases (normal: 0, acute: 22, chronic: 15), compared to previous 63 infection cases (normal: 0, acute: 31, chronic: 32). The ROC and AUC for IL-6 and CRP were revised in Figure 2. The new data was also shown in Table 1. Patient’s information including age, sex, laboratory data and microorganism or pathological findings was shown in Supplementary Table 1. All changes were highlighted and marked.
Point 2: There are already many studies dedicated to finding biomarkers, and many have studied the use of CRP and IL-6. The present paper only adds another method to detect IL6, it is not new as the authors published this in references 12 &13.
Response 2: Thanks for your suggestions. We agree that many papers have studied the role of IL-6 and CRP in wound infections. However, the role of systemic CRP in dertermination of wound infection is controverisal. CRP is a non-specific marker of inflammation and its potential as a diagnostic indicator for wound infection may be restricted depending on the presence of other inflammatory pathologies in individual patients. We have revised the non-specific role of CRP in the discussion part. Although there are studies to discuss IL-6 with wound infection, we are the first to investigate the cutoff levels of IL-6 and to analyze the pros and cons between IL-6 and CRP. In addition, we are the first to use a new technique by evaluationof tissue IL-6 for wound assessment.
Point 3: Details regarding patient and wound characteristics are missing: age, comorbidities, etc, and wound erythema, local heat, increased exudate or presence of microorganisms are mentioned but not reported. Other possible biomarkers for infection (such as procalcitonin) were not studied.
Response 3:
- Thanks for your suggestions. We have added age, sex and microorganism data or pathological findings of wounds in the Supplementary Table 1. Infected wounds must have the following clinical symptoms such as presence of wound erythema, local heat, increased exudate, and also have the laboratory or pathological finding of microorganisms or tissue infection.
- There are many biomarkers including IL-6 involved in wound infection. Our purpose was to develop a new IL-6 device for wound assessment. Our data also supported the role of IL-6 in descrimination of wound infection. Although the role of IL-6 was established in our study, we will discuss the role of other biomarkers such as procalcitonin in wound infection in the Discussion part for the possibility of development of a more potentially sensitive assay.
Point 4: Line 48 “other causative factors involved in wound inflammation”: what is meant by causative factors?
Response 4: thanks for comment, we have modified the inadequate expresison, “causative factors” into “biomarkers”.
Point 5: Line 57: IL-6 will not be useful for all types of wounds. For example, in burn wounds it will be highly increased in absence of infection. Also, IL6 can be expressed by a variety of cells after a multitude of stimuli. For example skin incision can increase IL6 levels for several hours.
Response 5: We agree your point about multi-function of IL-6 in clinical conditions. Despite of some extraordinary cases, our data support that tissue IL-6 was closedly related to infection status. We believe tissue IL-6 rather than systemic IL-6 can reflect real wound status. We have added to discuss the role of IL-6 in the Discussion part.
Point 6: CRP measurement is not included in M&M.
Response 6: A CRP blood test was used in clinical laboratory of our hospital using Beckman coulter image analyzer. We have added the procedure of CRP test in the Material and Methods.
Results
Point 7: Why include table 1 with information per sample? Why are some CRP results and many IL6 strip results missing?
Response 7:
- Thanks for your comments. The reason why we disclosed every patient’s information was to investigate the relationship between CRP and IL-6 in each patient. We also added additional patient’s profiles such as age, sex and microorganisms or pathological findings in the Supplementary Table 1.
-Yes, some patients in the normal tissue group do not have CRP data due to inconvenient acquisition of blood sample at the day of tissue harvested. In general, CRP should maintain in the baseline level in healthy subjects without external stimulation or associated diseases.
-Due to limited sample fluids, we study IL-6 concentrations in 15 cases using IL-6 test strip. Although only 15 cases are tested by IL-6 test strip, the similar results between conventional ELISA and new technique are shown in our study. We will continue to collect more cases to verify the validity of IL-6 test strip.
Point 8: Important information is missing: from which patients were multiple samples taken? 15 patients had serial samples for persistent infection, but which patients and was this repeated measurement taken into account?
Response 8: We have added patient ID in each sample to clarify the correlation of patient with sample in the Supplementary Table 1. Persistent infection was considered to repeat sampling every 5 to 7 days if infection is not controlled.
Point 9: There is a significant difference in IL6 between acute and chronic wounds, but all wounds were marked as infected. This indicates that IL6 cannot be used as a marker for infection. Why were CRP results of 2 normal samples >100 pg/ml?
Response 9:
- Thanks for your comment. I have reviewed our data and redefined wound infection as “ presence of wound erythema, local heat, increased exudate, and presence of microor-ganisms within a wound by laboratory bacteria examination or pathological finding of tissue infection. After evaluation, 37 cases were enrolled in the infection group cases (normal: 0, acute: 22, chronic: 15), compared to previous 63 infection cases (normal: 0, acute: 31, chronic: 32). A prognostic value of IL-6 for wound infection was still observed according to revised data.
- These 2 patients presented with acitve cancer status. Systemic CRP remained in a higher levels when normal tissue was harvested during regular surgery. This result further suggested a non-specific role of systemic CRP in determination of local wound infection.
Point 10: 3.2 Why is CRP “unreliable for determining local wound infection status” even when it is sensitive? There is more literature on the limited use of CRP for determining infection. Why not test other biomarkers or CRP/IL6 in plasma?
Response 10:
-1. We agree the concept that systemic CRP is sensitive for systemic infection and other disease such as cardiac disease. However, local wound infection may not stimulate systemic response such as elevated CRP, WBC or other markers. We may consider to study the relationship between tissue CRP and clinical wound infection.
-2. The role of IL-6 in wound infection is well described (Ref 9, 10). The prognositc value of IL-6 for determination of wound infection is superior to systemic CRP in our study. We are the first to develop IL-6 test strip to determine wound infection. If more sensitive and specific marker is identified, we may consider to develop a new test strip for wound assessment.
Point 11: Line 150-1 “suggesting a promising role for IL-6 as a biomarker for evaluating wound infection”: this is not based on presented results. All acute and chronic wounds were infected, all normal wounds were not. IL6 is not useful for infection in chronic wounds as those values were significantly lower than in acute samples. Non-infected acute wounds were not included, so it is not possible to draw a conclusion on infection status without additional info or samples.
Response 11: Thanks for your suggestions. We have revised our data to redefine wound infection in the previous description. After evaluation, infective cases have changed from 63 to 37 cases.
Point 12: What are the benefits of using strips over ELISA? For the strips a spectrum-based optical reader is needed.
Response 12: We have published many papers to show the advantages of paper-based ELISA technique in clinical diagnosis of many diseases. Paper-based IL-6 test strip technique had the advantages of economy, rapidity, tiny sample demands, and not inferior to conventional ELISA method. This paper-based technique required only 15 μL of reagent and an hour of processing, compared to the 550 μL and 7 h–8 h required for conventional plate ELISA. Additionally, conventional plate ELISA requires a plate reader and test-strip results can be recorded with a mobile phone. After dripping the sample fluid on the test zone, the picture of IL-6 on the test strip can be captured and activated by a mobile phone APP. The photo was scanned by the spectrum-based optical reader to acquire quantitative results. We have revised in the Discussion part.
Point 13: Lin 227-8: “our data also support the hypothesis that differential IL-6 levels could discriminate disease severity.” This is not shown here.
Response 13: Thanks for your comment. We have deleted this sentence and revised in the Discussion part.
Point 14: The discussion should be improved and expanded.
Response 14: Thanks for your suggestion. We have revised and expanded content in the Discussion part.

Round 2
Reviewer 1 Report
Dear Authors, I have two things prior publication. Thank you.
Point 4: "- We have tested 15 cases using paper-based IL-6 test strip in wound assessment. Due to limited samples, we can not analyze all samples with both techniques. However, a good correlation of test strip and conventional ELISA was observed in our study. We will collect more cases to verify this hypothesis. "
This must be described as a limitation in the revised version.
Response 6: I would rather not recommend to do local antibiotics. This is neither useful, nor recommended. If you have written sth.like this, I would recommend only to write systemic abx. Thank you.
Author Response
Response to Reviewer 1 Comments
Point 1: Point 4: "- We have tested 15 cases using paper-based IL-6 test strip in wound assessment. Due to limited samples, we can not analyze all samples with both techniques. However, a good correlation of test strip and conventional ELISA was observed in our study. We will collect more cases to verify this hypothesis. "
This must be described as a limitation in the revised version.
Response 1: thanks for the suggestion, we have added the limitation in the Discussion Part. (Line 293-295)
Point 2: Response 6: I would rather not recommend to do local antibiotics. This is neither useful, nor recommended. If you have written sth.like this, I would recommend only to write systemic abx. Thank you.
Response 2: thanks for the suggestion, we agree with your view in the role of local Abx. We have modified our related context.

Reviewer 3 Report
Thanks for clarifying all the mentioned issues.
I still wonder whether this IL6 strip is suitable for testing chronic wound fluid. At present only 3 samples were tested with this strip. On what grounds can you predict the strip will perform well enough on wound fluid? It would be good if more cases could be included in this publication.
typo line 268: consisted > consistent?
How does this research impact patient well being or wound care? the strip techniques is faster (and cheaper) than ELISA but does that benefit or change wound care? Will treatment be different or done at an earlier stage? That should be included.
Author Response
Point 1: I still wonder whether this IL6 strip is suitable for testing chronic wound fluid. At present only 3 samples were tested with this strip. On what grounds can you predict the strip will perform well enough on wound fluid? It would be good if more cases could be included in this publication.
Response 1: Thanks for your suggestions. Althogh there are some different conditions between acute and chronic wound fluids, even between different types of acute wounds, the paper-based IL-6 test strip is disigned with specific antibody targeting at IL-6 in wound tissue fluid. In this study, we confirmed our extraction protocol of wound tissue fluid that was suitable for IL-6 study and the positive relationship between paper-based IL-6 test strip results and conventional IL-6 ELISA results (Figure 3B&C) without considering and sub-classifying of wound types. Indeed, more chronic wound data could decrease such possible bias exist between acute and chronic wounds, which would definitely be included in our futher study.
Point 2: typo line 268: consisted > consistent?
Response 2: Thanks for your suggestions, we have revise the error!
Point 3: How does this research impact patient well being or wound care? the strip techniques is faster (and cheaper) than ELISA but does that benefit or change wound care? Will treatment be different or done at an earlier stage? That should be included.
Response 3: Thanks for your comments. We have included the benefit of IL-6 test strip in Discussion Part. IL-6 is known as a pro-inflammatory cytokine, and in this study, we found that significantly higher level of IL-6 in acute infected wound than normal and non-infected wound. If wound IL-6 is high, which means possibility of severe infection status of the wound, aggressive debridement and local/systemic anti-microbial agents should be used as soon as possible. Although conventional wound microorganism culture result could be a marker of infected wound, if takes at least 3-5 days for wound culture report and there are some microorganisms known as viable but non-culturable. Besides, assessment of IL-6 levels rather than microbial results may quantiate the infection severity of wounds according to the cutoff values. Like use of WBC or systemic CRP to evaluate systemic condition, we can follow the variation of tissue IL-6 levels to help assessment of wound infection status.
